# Evolution of Rheumatoid-Arthritis-Associated Interstitial Lung Disease in Patients Treated with JAK Inhibitors: A Retrospective Exploratory Study

**DOI:** 10.3390/jcm12030957

**Published:** 2023-01-26

**Authors:** Vincenzo Venerito, Andreina Manfredi, Antonio Carletto, Stefano Gentileschi, Fabiola Atzeni, Serena Guiducci, Marlea Lavista, Laura La Corte, Elisa Pedrollo, Arnaldo Scardapane, Caterina Tomassini, Bruno Frediani, Carlo Salvarani, Florenzo Iannone, Marco Sebastiani

**Affiliations:** 1Rheumatology Unit, Department of Precision and Regenerative Medicine—Ionian Area, University of Bari “Aldo Moro”, 70124 Bari, Italy; 2Rheumatology Unit, Azienda Ospedaliera Policlinico di Modena, University of Modena and Reggio Emilia, 41124 Modena, Italy; 3Rheumatology Unit, University of Verona, 37126 Verona, Italy; 4Rheumatology Unit, Azienda Ospedaliero-Universitaria Senese, Università Degli Studi di Siena, 53100 Siena, Italy; 5Rheumatology Unit, University of Messina, 98125 Messina, Italy; 6Rheumatology Unit, University of Florence, 50139 Firenze, Italy; 7Radiology Unit, University of Bari “Aldo Moro”, 70124 Bari, Italy; 8Division of Rheumatology, Azienda USL-IRCCS Reggio Emilia, 42421 Reggio Emilia, Italy

**Keywords:** rheumatoid arthritis, interstitial lung disease, JAK inhibitors

## Abstract

Background: The aim of this multicenter retrospective study was to investigate the effectiveness and safety of the available JAK-inhibitors (JAKi) in patients with rheumatoid arthritis (RA) and interstitial lung disease (ILD). Methods: We retrospectively analyzed patients with classified RA and RA-ILD undergoing JAKi in 6 Italian tertiary centers from April 2018 to June 2022. We included patients with at least 6 months of active therapy and one high-resolution chest tomography (HRCT) carried out within 3 months of the start of JAKi treatment. The HRCT was then compared to the most recent one carried out within 3 months before the last available follow-up appointment. We also kept track of the pulmonary function tests. Results: We included 43 patients with RA-ILD and 23 males (53.48%) with a median age (interquartile range, IQR) of 68.87 (61.46–75.78) treated with JAKi. The median follow-up was 19.1 months (11.03–34.43). The forced vital capacity remained stable in 22/28 (78.57%) patients, improved in 3/28 (10.71%) and worsened in 3/28 (10.71%). The diffusing capacity of lung for carbon monoxide showed a similar trend, remaining stable in 18/25 (72%) patients, improving in 2/25 (8%) and worsening in 5/25 (20%). The HRCT remained stable in 37/43 (86.05) cases, worsened in 4/43 (9.30%) and improved in the last 2 (4.65%). Discussion: This study suggests that JAKi therapy might be a safe therapeutic option for patients with RA-ILD in a short-term follow-up.

## 1. Introduction

Rheumatoid arthritis (RA) is a chronic autoimmune disease characterized by joint inflammation and destruction, with an incidence of 0.5–1% in the adult population of Western countries [1,2]. In addition to joint involvement, extra-articular manifestations of RA can affect several organs and systems [3].

Approximately 10% of the RA population develops a clinically significant ILD responsible for the decreased quality of life and progressive chronic disability. ILD is also connected to 10–20% of all mortality associated with the disease, with a mean survival of 5–8 years [3].

The treatment to adopt when ILD is detected in RA patients has always been a matter of debate. The management and treatment of RA-ILD is challenging because there is still little information available on this topic, and the main literature comes from observational studies. No clinical trials have been dedicated to this topic; however, its consideration is increasing in guidelines [4]. Current therapies for the treatment of RA are being extensively evaluated worldwide in order to determine their effect on the lung and on RA-ILD patients. The introduction of biotechnological disease-modifying antirheumatic drugs (bDMARDs) and, more recently, of Janus-kinases inhibitors (JAKi) has changed the course of RA, markedly improving the control of synovitis and consequently reducing joint destruction and physical disability [5]. We have also recently underlined the “protective” effect of abatacept [6] and tocilizumab [7], leading to a substantial stability of ILD, both assessed with high-resolution computed tomography (HRCT) and with pulmonary function tests (PFT). Rituximab has also been suggested for treating patients with RA-ILD, but its high risk of infections, when compared to abatacept, undermines its viability in such a complex patient subset [8].

There are relatively limited data on the use of JAKi in patients with RA-associated ILD. A mouse model demonstrated that tofacitinib (TOF) reduced arthritis and ILD by enhancing the expansion of myeloid-derived suppressor cells. TOF remarkably suppressed the progression of ILD in mice compared to controls [9]. A single-centre retrospective study from Tardella et al. [8] also reported scarce or null fibrosis progression in HRCT assessment in patients with RA-ILD treated with JAKi.

In this study, we aimed to investigate the effectiveness and safety of the available JAKi in patients with RA-ILD in a multicenter retrospective cohort with a central DICOM review performed by an expert radiologist.

## 2. Material and Methods

We retrospectively analyzed patients with RA classified according to the current criteria [1] and RA-ILD undergoing JAKi in 6 Italian tertiary centers from April 2018 to June 2022.

The different patterns of interstitial lung involvement defined by chest HRCT were classified according to the standardized criteria of the American Thoracic Society/European Respiratory Society International Multidisciplinary Consensus Classification of the Idiopathic Interstitial Pneumonias [10]. Clinical and demographic characteristics were recorded, including RA and ILD disease duration, rheumatoid factor (RF) and anti-cyclic citrullinated peptide antibody (ACPA) status.

We included patients with at least 6 months of active therapy and at least one HRCT carried out within 3 months before the start of the JAKi treatment. This HRCT, together with the most recent HRCT, carried out within 3 months from the last available follow-up, was sent to the central DICOM and was reviewed by an expert radiologist. In particular, we defined ILD as “worsened” if the fibrosing features were greater than 10% compared to previous HRCT. We defined “stable” as those with progression or reduction in fibrosis of less than 10% and “improved” as those with a more than 10% reduction in ILD.

We kept track of the ILD patterns and eventual acute exacerbations. We also recorded pulmonary function tests (PFTs) corrected for age, gender, and height, carried out in the 3 months before the JAKi treatment started and at the last available follow-up. The results of PFTs were expressed as percentages of the predicted value for each parameter and corrected for age, gender and height. Pulmonary function was considered abnormal if forced vital capacity (FVC) was <80% of predicted values. Single-breath Diffusing Capacity of Lung for Carbon Monoxide (DLCO-SB) and DLCO adjusted by the alveolar volume (DLCO-VA) were used to assess gas transfer. For FVC, we defined “stable” those with lung function improvement or deterioration of less than 20% and “improved” those with a more than 20% increase. Conversely, patients were defined as “worsened” if FVC deterioration was greater than 20%. For DLCO, an improvement or deterioration of 15% had to be observed to classify patients as improved or worsened, respectively.

Mann–Whitney U-test was used to test the difference between DLCO and FVC at either JAKi baseline or at the last available follow-up. Fisher’s exact test was used in contingency tables. Kaplan–Meier estimates were used to investigate the survival functions of the different JAKi. No imputation for missing data was carried out. This study received local ethical committee approval (Comitato Etico dell’Area Vasta Emilia Nord 109/2019/OSS/AOUMO). All living patients gave written informed consent.

## 3. Results

We included 43 patients with RA-ILD, 23 were male (53.48%) and the median age (Interquartile Range, IQR) was 68.87 (61.46–75.78); they were treated with JAKi.

For all patients, HRCT was available in the previous 3 months before the beginning and at the end of the therapy with JAKi, whereas FVC was available in 27/43 (62.79%) patients and DLCO in 30/43 (69.76%). Repeated FVC measures over time were available for 28 patients (65.11%), whereas DLCO was only available for 25 (58.13%).

Baricitinib (BAR) was prescribed in 28 out of 43 patients (65.12%) and TOF in 9/43 (20.93%), whereas either filgotinib (FIL) or upadacitinib (UPA) were prescribed in 3 patients (6.98%, respectively).

All patients were naïve to anti-fibrotic drugs and none of them received concomitant nintedanib during the observation period.

The baseline characteristics of RA-ILD patients included in the study are summarized in Table 1.

The median follow-up was 19.1 months (interquartile range (IQR) 11.03–34.43). Thirty-eight were positive for RF (88.37%) and all but eight for ACPA (81.40%). A usual interstitial pneumonia (UIP) pattern was described in 25/43 (58.14%) patients, whereas a non-specific interstitial pneumonia (NSIP) pattern was identified in five other patients (11.62%). Lymphocytic interstitial pneumonia (LIP) was present in 2/43 patients (4.65%). In one patient only, a combined pattern of pulmonary fibrosis and emphysema (CPFE) was recorded (2.33%). Ten out of forty-three patients had an indeterminate pattern (23.26%). All patients experienced therapies with synthetic or biologic DMARD before JAKi. Thirty-two out of forty-three (74.41%) patients had previously been treated with methotrexate (MTX), and three out of forty-three (6.97%) with leflunomide. With regard to biologic DMARDs, 19/43 (44.19%) patients experienced therapy with TNF inhibitors (TNFi), 12/43 (27.91%) with rituximab, 16/43 (38.10%) with abatacept and 13/43 (30.23%) with tocilizumab. All patients were JAKi-naïve. Only 16 out of 43 patients took JAKi with MTX combotherapy (38.10%); on the other hand, 26/43 (60.47%) patients were on glucocorticoids at JAKi baseline. The evolution of lung function and radiology is summarized in Figure 1.

After a median follow-up of 19.1 months, FVC remained stable in 22/28 (78.57%) patients, and it improved in 3/28 (10.71%) and worsened in 3/28 (10.71%). At the end of the follow-up, the mean FVC did not change significantly from the JAKi treatment baseline (mean difference from baseline 3.39% ± 11.18, *p* = 0.12). DLCO showed a similar trend, remaining stable in 18/25 (72%) patients, improving in 2/25 (8%) and worsening in 5/25 (20%). A slight but statistically significant decrease was recorded at the end of the follow-up for DLCO (mean difference from baseline 3.44% ± 7.18, *p* = 0.02). HRCT remained stable in 37/43 (86.05%) cases, worsened in 4/43 (9.30%) and improved in the other 2 (4.65%). The worsened HRCT was recorded in two patients with UIP and two with NSIP, respectively. In contrast, the improved HRCT was observed in a patient with NSIP and one with an indeterminate pattern. Table 2 summarizes the HRCT evolution of RA-ILD in the different treatment groups. No difference in terms of expected frequency was found in the contingency table (*p* for Fisher’s exact test = 1.00)

Similarly, no improvement or deterioration was seen in the HRCT and PFTs between those with MTX combotherapy and patients with JAKi monotherapy. Between the follow-up periods, JAKi were withdrawn in 21 out of 43 patients (48.84%): in 15 out of 21 (71.42%) patients this was due to a loss of effectiveness in terms of articular involvement, whereas in 3 out of 21 patients (14.28%) this was due to JAKi-unrelated adverse events.

Two withdrawals (one death) were recorded for acute exacerbations of ILD out of forty-three patients (9.52%), all in individuals on BAR. One patient’s treatment (on BAR) was discontinued due to a serious infection (4.76%). The causes of discontinuation and adverse events have also been reported in Table 3.

The median follow-up time was different between non-selective and selective JAKi, reflecting the recent availability of UPA and FIL in Italy. Patients on BAR had a median follow-up of 20.7 months (IQR 13.41–34.16) and those on TOF of 22.13 months (IQR 9.96–36.53). On the other hand, those on UPA had a median follow-up of 15.13 months (IQR 10–15.13), whereas patients on FIL had a median follow-up of 7 months (IQR 6.1–8). Restricted mean survival time for BAR was 32.78 months (95%CI 26.32–39.24), 34.71 months (95%CI 19.38–50.03) for TOF, 7.57 months (98%CI 7.57–7.57) for FIL and 15.13 (95%CI 15.13–15.13) for UPA (underestimated because of the censoring of the largest observed analysis time). Figure 2 shows our cohort’s survival function and the Kaplan–Meier separate estimates for each JAKi. Figure 3 representatives HRCTs.

## 4. Discussion

In our study, we evaluated the effectiveness and safety of JAKi in a retrospective cohort of patients affected with RA-ILD. The treatment of such patients is challenging due to the possible role of bDMARDs in the progression of the disease and the development of acute exacerbation.

Several authors described a possible effect of all TNFi in the new onset or exacerbation of RA-ILD [11,12,13]. Perez-Alvarez et al. and the British Society of Rheumatology have historically cautioned against prescribing TNFi to patients with RA-ILD for the supposed increased risk of exacerbation of ILD [13,14].

Conversely, there is increasing evidence that rituximab, tocilizumab and abatacept are associated with better patient survival; substantial PFTs stability; and little or no HRCT deterioration [6,7,15,16] in patients with RA-ILD.

Scarce data have been published on the clinical and radiological outcomes of RA-ILD patients on JAKi. Aside from anecdotal case reports [17], JAK inhibition had been considered relatively safe, since the pooled incidence of new-onset ILD, following tofacitinib treatment, was 0.18 per 100 patients/year in a pooled analysis of phases 1,2,3, 3a, 4 and long-term extension studies [18].

Recently, the impact of the first-generation JAKi on the short-term evolution of RA-ILD was evaluated in a small Italian cohort of 31 patients. In the cohort, 18 patients were on BAR, whereas 13 were on TOF. The extension of ILD was assessed with a computer-aided method able to estimate the percentage of fibrosis in the HRCT. At the 18 month follow-up, 5/31 (16.1%) patients showed a HRCT deterioration (i.e., ≥15% progression of fibrosis from baseline HRCT), 20/31 (64.5%) were considered stable (i.e., progression or reduction in fibrosis <15%) whereas 6/31 (19.4%) patients showed an HRCT improvement (i.e., ≥ 15% reduction in fibrosis) [8].

Mean FVC and DLCO remained stable at the 18-month follow-up of JAKi therapy.

Such data are somewhat concordant with results from our cohort, where most patients, about 90%, experienced no ILD progression at the follow-up HRCT. Additionally, our criteria was slightly stricter than the one in the above report. We considered a more conservative cut-off of 10% fibrosis progression to define an RA-ILD worsening.

Consistent with Tardella et al. [8], we also found that patients on JAKi had stable FVC during the observation period.

Moreover, although we observed stability or improvement of DLCO in more than two thirds of our cohort, the mean DLCO at the end of follow-up was significantly reduced. Nevertheless, one should consider that a mean decrease of about 3 percentage points over a median of 19.1 months of follow-up might not be clinically significant.

About a third of patients were treated with combination therapy, including MTX. Despite inconclusive results, recent data have suggested the safety of MTX in RA-ILD patients [19]. The American College of Rheumatology (ACR) [20] stated that clinicians (rheumatologists and pulmonologists) should prioritize any alternative option rather than accepting any additional risk of lung toxicity; therefore, the ACR conditionally recommends MTX over alternative DMARDs for the treatment of RA-ILD patients.

Less than half of our cohort discontinued JAKi by the end of the observation period. These data are somewhat consistent with the recent report by Kalyoncu et al. [21]. In this report, Kalyoncu et al. observed a retention rate of 42.6% over a median follow-up of 15 months (IQR 9–32) in a cohort of 46 patients with RA-ILD on TOF. The latter was not significantly different from the retention rate observed in patients on TOF without RA-ILD by the same authors (44.3%, median follow-up 11 (IQR 4–24 months)). In comparison, our cohort was different. It included patients with different JAKi, either selective or non-selective.

We found that discontinuation happened mainly due to the loss of effectiveness on articular involvement, and only one serious infection leading to BAR discontinuation was recorded. Finally, we observed 2 acute exacerbations of RA-ILD out of 43 patients (4.65%), 1 of which was lethal.

Acute exacerbations may occur, with a reported incidence of 5.77 patients/year in those with autoimmune ILD. A study by Manfredi et al. [22,23] observed 2 cases of acute exacerbations (10.52%) over a mean follow-up of 23.9 ± 10.9 months in a prospective cohort of 19 patients with RA-ILD. The two patients with acute exacerbation were on etanercept and methotrexate, respectively. That is to say, further studies are needed on prospective cohorts in order to investigate whether JAKi therapy might prevent acute exacerbations in patients with RA-ILD.

Our study has several limitations. First, we must acknowledge the low sample size and short follow-up time frame, especially for selective JAKi, such as FIL and UPA. Moreover, the retrospective design of the study did not allow us to record some meaningful data, such as the patients’ disease activity either at baseline or at the end of the observation period. Recently, an ORAL surveillance study [24] raised some concerns about the safety of JAKi, particularly in connection with the risk of lung cancer. On the other hand, ILD represents per se one of the most significant risk factors for lung cancer; therefore, patients with ILD should be carefully monitored for this complication. In the absence of a specific post hoc analysis on the ORAL surveillance study [24], any definitive conclusion can be made on the long-term risk of lung cancer in RA-ILD patients treated with JAKi. Since international scientific societies have not yet definitively expressed the possible use of MTX, this could have suggested a larger use of JAKi as monotherapy and influenced the choice of specific JAKi.

## 5. Conclusions

Our study suggest that JAKi therapy might be a safe therapeutic option for patients with RA-ILD, leading to a substantial stability of ILD at HRCT and possibly preventing PFTs deterioration, at least in a short-term follow-up.

Further research on a larger prospective cohort with long-term follow-up is needed to corroborate our results.

## Figures and Tables

**Figure 1 jcm-12-00957-f001:**
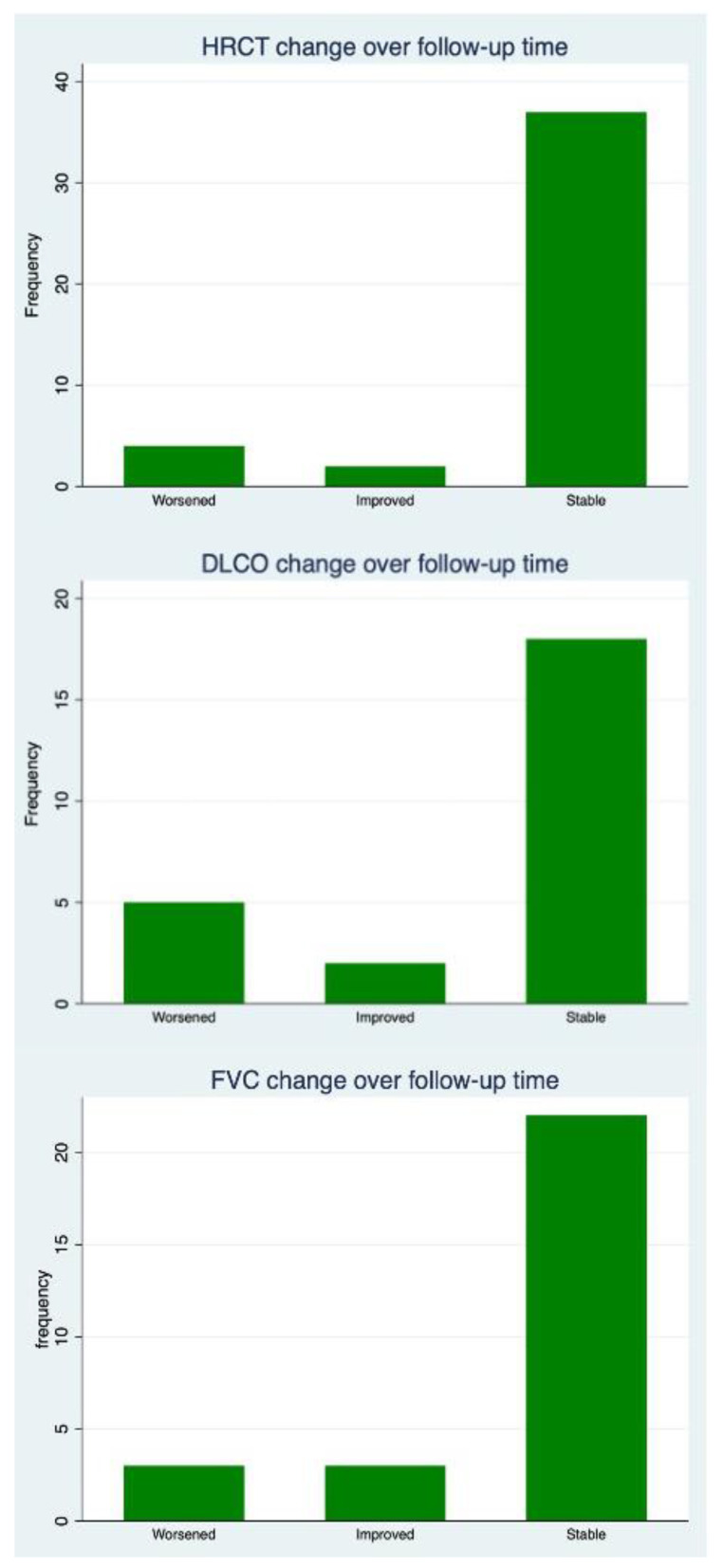
Pulmonary function tests and high-resolution computed tomography (HRCT) evolution. Upper Panel: ILD evolution at HRCT, Middle Panel: diffusing capacity of lung for carbon monoxide (DLCO). Lower Panel: forced vital capacity (FVC).

**Figure 2 jcm-12-00957-f002:**
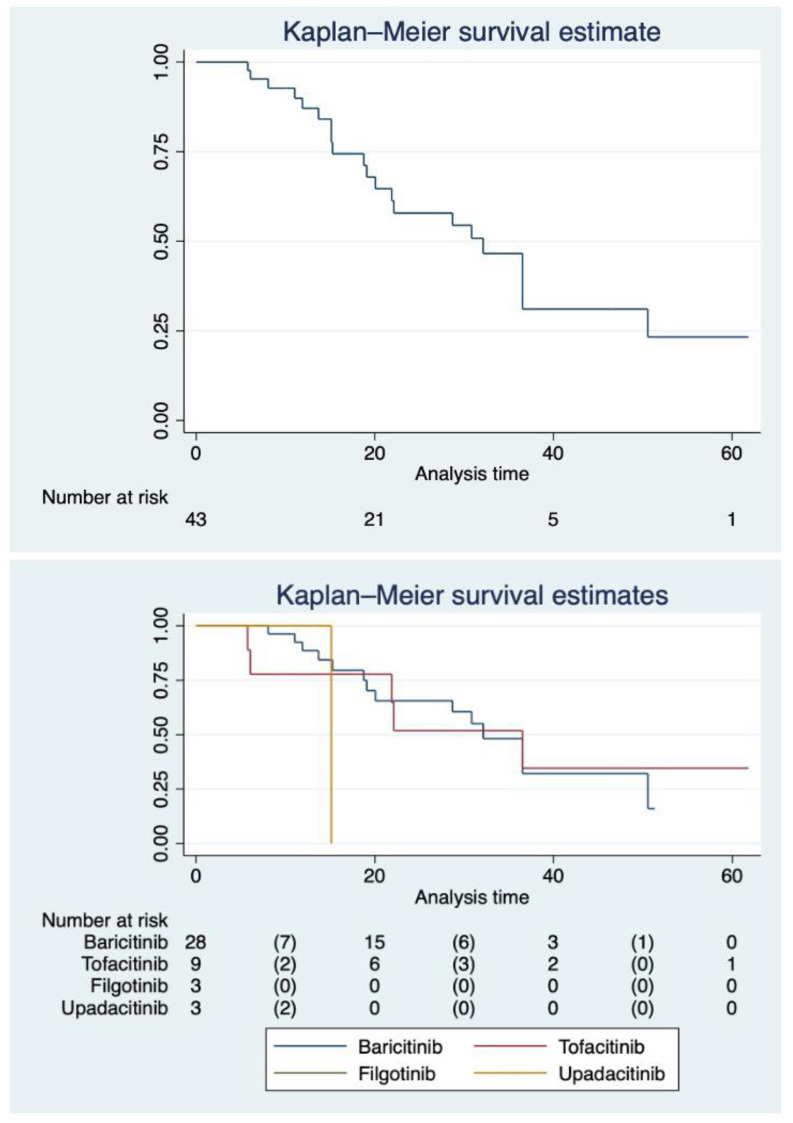
Survival function of patients with RA-ILD on JAKi. Upper panel: survival function in the whole cohort with the at-risk panel. Lower panel: survival function stratified by different JAKi with at-risk panel.

**Figure 3 jcm-12-00957-f003:**
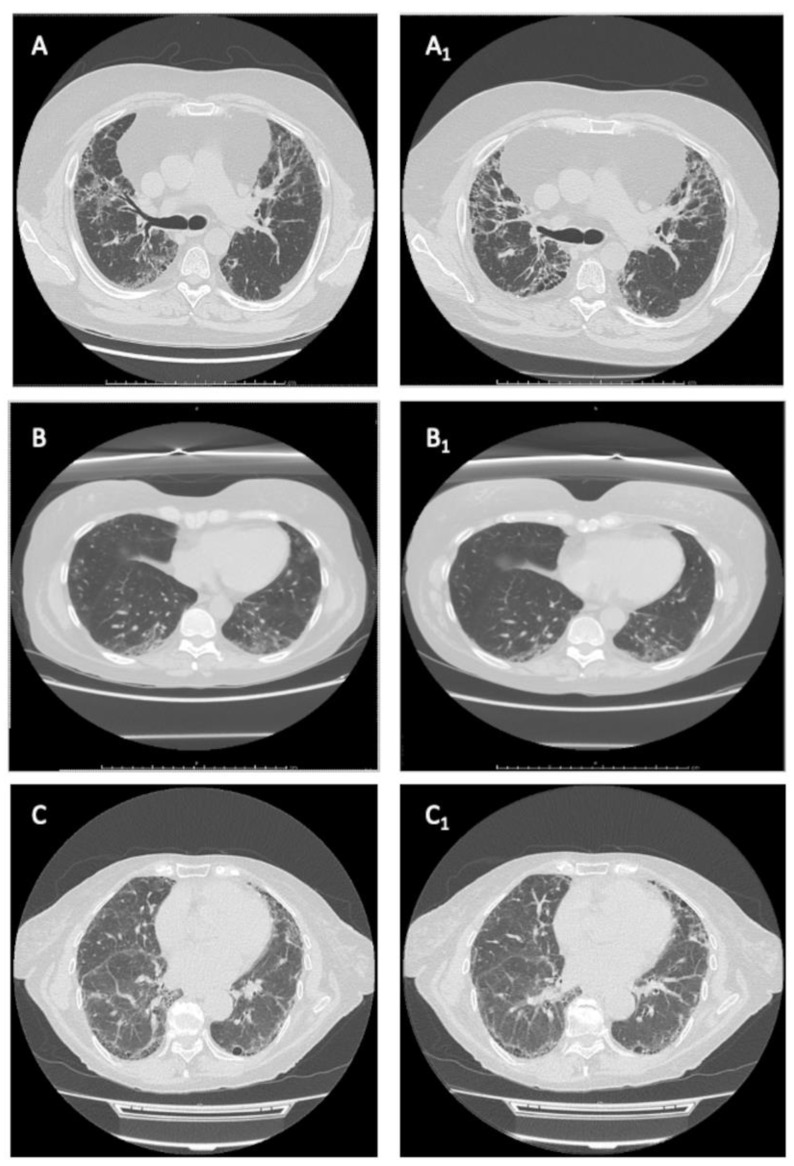
Representative HRCTs. (**A**,**A_1_**) series: a “worsened” RA-ILD with UPI pattern in a patient treated with baricitinib. (**B**,**B_1_**) series: an “improved” RA-ILD with indeterminate pattern in a patient treated with baricitinib. (**C**,**C_1_**) series: a “stable” RA-ILD with UIP pattern in a patient treated with baricitinib.

**Table 1 jcm-12-00957-t001:** Patient characteristics at treatment baseline.

	Av Obs.	Baseline
Age, years, median (IQR)	43	68.87 (61.46–75.78)
Male, *n* (%)	43	23 (53.48)
Disease Duration, years, median (IQR)	43	12.66 (7.61)
ILD duration, years, median (IQR)	43	5.55 (5.13)
Follow-up, months, median (IQR)	43	19.1 (14.92)
Rheumatoid factor positivity, *n* (%)	43	38 (88.37)
ACPA positivity, *n* (%)	43	35 (81.40)
HRCT pattern, *n* (%)	43	
UIP		25 (58.14)
NSIP		5 (11.62)
LIP		2 (4.65)
CPFE		1 (2.33)
Indeterminate		10 (23.26)
Baseline DLCO, mean (SD)	27	65.81 (16.92)
Baseline FVC, mean (SD)	30	88.76 (24.03)
Prescribed JAKi, *n*/%)	43	
Baricitinib		28 (65.12)
Filgotinib		3 (6.98)
Tofacitinib		9 (20.93)
Upadacitinib		3 (6.98)
Use of DMARD before JAKi, *n* (%)	43	
Methotrexate		32 (74.41)
Leflunomide		3 (6.97)
TNFalpha inhibitors		19 (44.19)
Rituximab		12 (27.91)
Abatacept		16 (38.10)
Tocilizumab		13 (30.23)
JAKi + Methotrexate, *n* (%)	43	16 (37.21)
Glucorticoids *n* (%)	43	26 (60.47)

Abbreviations: ACPA: anti-citrullinated peptide antibodies; CPFE: combined pattern pulmonary fibrosis and emphysema; DLCO: diffusing capacity of lung for carbon monoxide; DMARD: disease-modifying anti-rheumatic drug; FVC: forced vital capacity; JAKi: Janus kinase inhibitors; IQR: interquartile range; ILD: interstitial lung disease; LIP: lymphocytic interstitial pneumonia; NSIP: non-specific interstitial pneumonia; UIP: usual interstitial pneumonia; TNF: tumor necrosis factor.

**Table 2 jcm-12-00957-t002:** HRCT evolution in different JAKi groups.

Outcome n (Column%; Row%)	Baricitinib (n.28)	Tofacitinib (n.9)	Filgotinib (n.3)	Upadacitinib (n.3)	Total (n.43)
Improved	2 (7.14; 100)	0	0	0	2 (4.65; 100)
Stable	23 (82.14; 62.16)	8 (88.89; 21.62)	3 (100; 8.11)	3 (100; 8.11)	37 (86.05; 100)
Worsened	2 (10.71; 75)	1 (11.11; 25)	0	0	4 (9.30; 100)
Total	28 (100; 68.12)	9 (100; 20.93)	3 (100; 6.98)	3 (100; 6.98)	43 (100; 100)

*p* for Fisher’s exact test = 1.0; No differences were recorded according to the duration of follow-up or the previous therapies.

**Table 3 jcm-12-00957-t003:** Causes of treatment discontinuation and adverse events.

	Available Observations	
Treatment Discontinuations, *n* (%)	43	21 (48.84)
Acute Exacerbations, *n* (%)	21	2 (9.52)
Loss of Effectiveness on Joint Involvement, *n* (%)	21	15 (71.42)
Death, *n* (%)	21	1 (4.76)
Serious infection (urinary tract, *n* (%)	21	1 (4.76)
Other, *n* (%)	21	3 (12.28)

## Data Availability

Data are available upon request to the corresponding author.

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
