# Peer review of "Evolution of Rheumatoid-Arthritis-Associated Interstitial Lung Disease in Patients Treated with JAK Inhibitors: A Retrospective Exploratory Study"

_jcm, 2023, doi:10.3390/jcm12030957_

Round 1
Reviewer 1 Report
Dear Editor,
The authors' aim is to evaluate the effectiveness and safety of the available JAK inhibitors in patients with RA and Interstitial Lung Disease.
Retrospective design, a small number of patients (quite a few), lack of control groups (vs DMARDs or biologicals, etc), and also lack of information about disease activity are the major and most important factors limiting the study. It seems difficult to draw any conclusions about the safety and efficacy of JAK inhibitors due to the short follow-up period and the lack of a detailed description of comorbidities or cardiovascular risks in geriatric patients (due to increasing concerns) in this low-power study.
Best Regards
Author Response
The authors' aim is to evaluate the effectiveness and safety of the available JAK inhibitors in patients with RA and Interstitial Lung Disease. Retrospective design, a small number of patients (quite a few), lack of control groups (vs DMARDs or biologicals, etc), and also lack of information about disease activity are the major and most important factors limiting the study. It seems difficult to draw any conclusions about the safety and efficacy of JAK inhibitors due to the short follow-up period and the lack of a detailed description of comorbidities or cardiovascular risks in geriatric patients (due to increasing concerns) in this low-power study.
We would like to thank the reviewer for her/his constructive insights and for explaining the intrinsic limitations of retrospective study design in clinical research.
In so rare conditions as RA-ILD, multicentric observational retrospective studies have been widely used to evaluate treatment effects (Fernández-Díaz C, et al. Semin Arthritis Rheum. 2018;48:22-27; Manfredi A,et al. Intern Med J. 2020 Sep;50(9):1085-1090). We agree with the referee that this is a limit of these kind of studies; on the other hand, the possibility to develop prospective randomized trial is very difficult, as showed by the recent limitation in enrollment of RELIEF and TRAIL1 studies (Behr J, et al. Lancet Respir Med. 2021;9:476-486; Solomon J et al, Lancet Resp Med, 2022). Of note, our cohort of RA-ILD patients on JAKi (n.43) is one of the most representative in the literature after the one from the Turkish registry described by Kalyoncu et al. (i.e. n.47 patients on TOF) (Kalyoncu U, et al. Clin Exp Rheumatol. 2022;40:2071-2077). We agree that prospective studies would fit better the assessment of JAKi effectiveness and safety for which ORAL surveillance raised concerns, but we aimed to investigate only the tomographic evolution and eventual PFTs deterioration in such patients.
To highlight the points that the referee has correctly addressed, we have furtherly expanded the discussion about the limits of our study
Reviewer 2 Report
Review of “Evolution of Rheumatoid Arthritis-associated Interstitial Lung 2 Disease in patients treated with JAK inhibitors: a multicenter 3 retrospective study.”
The paper by Venrito et al explores the safety and effectiveness of JAK-inhibitors in RA patients with ILD.
Major comments:
- The authors claim to investigate safety and effectiveness but only data on adverse events are given in the discussion. Results of AE should be given in a table when safety is investigated. Also in the discussion it is mentioned that less than half of the cohort discontinued JAKi
- Table 1 should be shown in the manuscript before other results.
- Figure 1 is unclear in presentation and also upper and middle panel have the same name.
- The authors show very little results. They should show the results of ILD measures in different JAKi, and also related to the different forms of ILD. Moreover it is not explained why data are shown of 28 patients for FVC, and in 25 for DLCO. Only for HRCT all patient data are given, however not related to JAKi used.
Author Response
Q. The paper by Venerito et al explores the safety and effectiveness of JAK-inhibitors in RA patients with ILD.
Major comments:
The authors claim to investigate safety and effectiveness but only data on adverse events are given in the discussion. Results of AE should be given in a table when safety is investigated. Also in the discussion it is mentioned that less than half of the cohort discontinued JAKi
A. We would like to thank the reviewer for her/his suggestions. We summarized AE and causes of discontinuation in table 3 and we added Kaplan-Meir Survival estimates in Figure 2.
Q. Table 1 should be shown in the manuscript before other results.
A.Thank you. Amended.
Q. Figure 1 is unclear in presentation and also upper and middle panel have the same name.
A.Thank you. Amended.
Q. The authors show very little results. They should show the results of ILD measures in different JAKi, and also related to the different forms of ILD. Moreover, it is not explained why data are shown of 28 patients for FVC, and in 25 for DLCO. Only for HRCT all patient data are given, however not related to JAKi used.
A.We would like to thank the reviewer for her/his suggestions. Unfortunately, repeated measures were not available for some patients and consequently were not considered for paired analysis. We acknowledged it in the revised manuscript. Furthermore, we added the tomographic evolution stratified for treatment groups in the new table 2. The ILD patterns that worsened or improved had been included in results.
Reviewer 3 Report
Comments to Authors:
The authors investigated the effectiveness and safety of the available JAK-inhibitors (JAKi) in patients with Rheumatoid Arthritis(RA) and Interstitial Lung Disease(ILD) in a multi-24 center retrospective study. They concluded that JAKi therapy might be a safe therapeutic option for patients with RA-ILD. I think this study is interesting and important because the management and treatment of RA-ILD are challenging due to little information available on this topic and relatively limited data on using JAKi in patients with RA-associated ILD. However, some points should be added and addressed for this study to be further informative.
My primary concern is the lack of summary tables. Therefore, it is hard to understand the result section.
1. Comparisons of characteristics among “worsened”, “stable” and “improved” groups should be summarized in a table and further discussed to clarify possible differences.
2. Regarding drug safety, the frequency and details of infectious diseases during JAKi use should be mentioned because they are well-known possible side effects.
3. The information if the patients was on anti-fibrotic drug use, such as nintedanib should be added.
4. Representative chest CT images of each group, “worsened”, “stable”, and “improved”, should be added.
Author Response
Q. The authors investigated the effectiveness and safety of the available JAK-inhibitors (JAKi) in patients with Rheumatoid Arthritis(RA) and Interstitial Lung Disease(ILD) in a multi-24 center retrospective study. They concluded that JAKi therapy might be a safe therapeutic option for patients with RA-ILD. I think this study is interesting and important because the management and treatment of RA-ILD are challenging due to little information available on this topic and relatively limited data on using JAKi in patients with RA-associated ILD. However, some points should be added and addressed for this study to be further informative.
A. We would like to thank the reviewer for her/his appreciation. We would also like to apologize for the typos encountered due to the splicing our manuscript into the MDPI template. We amended them.
Q. My primary concern is the lack of summary tables. Therefore, it is hard to understand the result section. Comparisons of characteristics among “worsened”, “stable” and “improved” groups should be summarized in a table and further discussed to clarify possible differences.
A. We would like to thank the reviewer for her/his suggestions. We added the tomographic evolution stratified for treatment groups in the new table 2
Q. Regarding drug safety, the frequency and details of infectious diseases during JAKi use should be mentioned because they are well-known possible side effects.
A. We summarized AE and causes of discontinuation in table 3 and we added Kaplan-Meir Survival estimates in Figure 2
Q. The information if the patients was on anti-fibrotic drug use, such as nintedanib should be added.
A. Thank you. No patients received nintedanib. We added such information in the revised manuscript.
Q. Representative chest CT images of each group, “worsened”, “stable”, and “improved”, should be added.
A. We would like to thank the reviewer for her/his suggestion. In Figure 3 we added representative CT images for each group.
Round 2
Reviewer 3 Report
The contents seem much improved. P values should be added in Table 2, and the Italian language is likely to exist in the discussion, therefore the part should be corrected.
Author Response
Q.The contents seem much improved. P values should be added in Table 2, and the Italian language is likely to exist in the discussion, therefore the part should be corrected.
A. We would like to thank the reviewer. We had the Discussion revised by a native English speaker and amended the table. Please find attached the highlighted changes.
